# 🌍 GeoArena: An Open Platform for Benchmarking Large Vision-language Models on WorldWide Image Geolocalization

## Abstract

Image geolocalization aims to predict the geographic location of images captured anywhere on Earth, but its global nature presents significant challenges. Current evaluation methodologies suffer from two major limitations. First, static datasets: advanced approaches often rely on large vision-language models (LVLMs) to predict image locations, yet these models are frequently pretrained on the test datasets, compromising the accuracy of evaluating a model's actual geolocalization capability. Second, existing metrics primarily rely on exact geographic coordinates to assess predictions, which not only neglects the reasoning process but also raises privacy concerns when user-level location data is required. To address these issues, we propose **GeoArena**, *a first open platform for evaluating LVLMs on worldwide image geolocalization tasks, offering true in-the-wild and user-preference-based benchmarking*. GeoArena enables users to upload in-the-wild images for a more diverse evaluation corpus, and it leverages pairwise human judgments to determine which model output better aligns with human expectations. Our platform has been deployed online for three months, during which we collected over thousands voting records. Based on this data, we conduct a detailed analysis and establish a leaderboard of different LVLMs on the image geolocalization task. GeoArena has been open-sourced[1] to support future research.

## 1 Introduction

Image worldwide geolocalization is a highly challenging task that lies at the intersection of computer vision (Szeliski, 2022; He et al., 2016) and geographic artificial intelligence (Mai et al., 2022). It aims to pinpoint the exact location on Earth where a given image was taken, as illustrated in Figure 1. This task has significant application potential, such as in navigation, positioning, crime tracking, and disaster monitoring. Despite its broad utility, geolocalization remains a hard problem due to the enormous variability in visual appearances across the globe (Wilson et al., 2021; Vo et al., 2017), coupled with the need for fine-grained spatial reasoning.

Recently, advanced methods for image worldwide geolocalization have increasingly relied on powerful large vision-language models (LVLMs) to generate predictions (Vivanco Cepeda et al., 2023; Zhou et al., 2024; Jia et al., 2024). To assess the performance of different methods on geolocalization tasks, the research community has proposed a series of benchmark studies (Li et al., 2025b; Huang et al., 2025; Jay et al., 2025; Liu et al., 2024; Wang et al., 2024; Astruc et al., 2024). Without exception, these benchmarks are based on static datasets and evaluate model performance using ground-truth labels (i.e., by calculating the distance between the predicted GPS location and the ground-truth GPS), as shown in Table 1. This evaluation approach is efficient and requires relatively few resources. However, it has two notable limitations: (1) Static datasets: static test datasets are inevitably subject to data leakage, meaning that the test data may be included in the pretraining data of LVLMs; and (2) GPS-based evaluation: existing evaluations only consider the spatial distance between the final prediction and the ground truth, ignoring the model's reasoning process. This can lead to cases where models with flawed reasoning processes achieve higher scores by chance

---

[1] https://anonymous.4open.science/r/GeoArena-6EDE

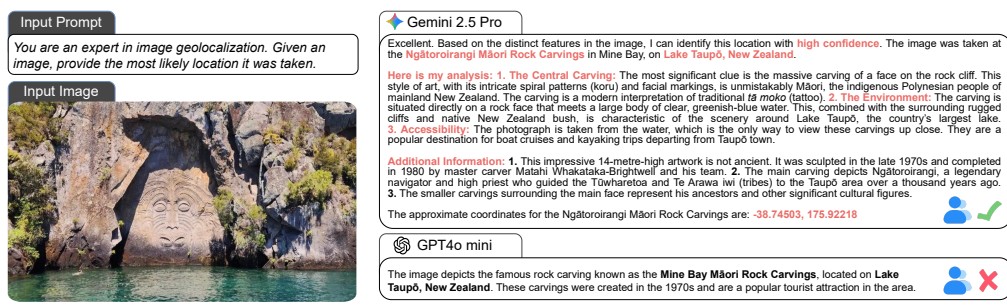

Figure 1: Example of geolocalization: identifying the Ngātoroirangi Māori Rock Carvings.

Table 1: Comparison with different benchmarks on different properties.

| Benchmarks | OSV-5M | LLMGeo | ETHAN | Location-Inference | FairLocator | IMAGEO-Bench | GeoArena |
|---|---|---|---|---|---|---|---|
| Conference | CVPR'24 | CVPR'24 | Arxiv'24 | AAAI'25 | Arxiv'25 | Arxiv'25 | - |
| Reference | Astruc et al. (2024) | Wang et al. (2024) | Liu et al. (2024) | Jay et al. (2025) | Huang et al. (2025) | Li et al. (2025b) | Ours |
| Evaluation | GPS | Country | GPS | GPS, Country, City | GPS, Street, City, Country, Continent | GPS, City, State, Country | User Preference |
| Dynamic Datasets | ✗ | ✗ | ✗ | ✗ | ✗ | ✗ | ✓ |
| User-Preference-Based | ✗ | ✗ | ✗ | ✗ | ✗ | ✗ | ✓ |

when they predict closer to the ground truth. Moreover, relying on exact geographic coordinates for evaluation raises privacy concerns, since it requires access to sensitive user-level location data. To address these limitations and develop a more effective evaluation approach, it is necessary to build a *dynamic and user-preference-based benchmarking platform that can capture the real-world challenges of image geolocalization*.

Designing such an evaluation platform is inherently challenging. It involves gathering a steady stream of diverse and representative user images that capture the variability of real-world conditions. Furthermore, the platform must support scalable and efficient evaluation pipelines capable of handling a wide range of model submissions. Finally, designing a reasonable model ranking system based on user preferences is also essential. These challenges highlight the need for careful design and robust infrastructure to create a meaningful and practical benchmarking platform for image geolocalization.

To this end, we introduce **GeoArena**, an open platform for benchmarking Large Vision-language Models on worldwide image geolocalization. Specifically, when a user enters GeoArena and submits an image for geolocalization, two anonymous models each generate a response indicating the predicted location. The user then votes on which response is more satisfactory. After collecting a large volume of voting data, we apply statistical methods to generate reliable rankings for all models. These rankings can serve as valuable references for users in the field, guiding them in selecting models that best align with geolocalization needs. In this way, GeoArena addresses two critical issues in current evaluation practices. First, GeoArena collects in-the-wild images contributed by real-world users, ensuring data diversity and dynamic updates, which help mitigate data leakage from static datasets. Second, GeoArena employs user preferences to assess the quality of model predictions, moving beyond sole reliance on GPS accuracy and mitigating the privacy risks associated with requiring exact user location data. Through these improvements, GeoArena establishes a new evaluation framework for image geolocalization that is more dynamic, privacy-preserving, and reflective of real-world user preferences. This approach bridges the gap between automated metrics and human-centered evaluation, providing a more robust and generalizable benchmark for the community.

GeoArena has been successfully deployed since June 2025, operating for over three months and collecting thousands of records. Our deployment has already revealed clear patterns: frontier systems like Gemini 2.5 dominate the leaderboard, while strong open-source families such as Qwen 2.5 and Gemma 3 are closing the gap. User preference analysis shows that longer and more structured responses are consistently favored, highlighting the importance of reasoning quality. We also find that top proprietary models align more closely with human judgment than open-source models, though noticeable gaps remain. These insights demonstrate GeoArena's ability to both benchmark models dynamically and uncover the factors that shape human evaluations in geolocalization. We will release all the collected voting data to support advancements in related areas, such as reward modeling (Zhong

et al., 2025) and geographic foundation models (Mai et al., 2024). Our key contributions can be summarized as follows:

1. We develop GeoArena, the first dynamic, user-preference-based open platform for addressing long-standing issues in geolocalization evaluation, including static-dataset leakage, missing reasoning assessment, and privacy concerns.

2. We conduct a comprehensive analysis of the collected user inputs and voting data to demonstrate the reliability and capabilities of GeoArena.

3. We publicly release the collected prompts, images, and voting data to support research and development in related fields such as reward modeling and geographic foundation models.

## 2 RELATED WORK

**Worldwide Image Geolocalization.** Worldwide image geolocalization is an interdisciplinary task that bridges geography and computer science, involving GeoAI (Janowicz et al., 2020), spatial data mining (Wang et al., 2020), and multi-modal modeling (Wang et al., 2023). In recent years, thanks to the strong world knowledge and visual understanding capabilities of Large Vision-Language Models (LVLMs), image geolocalization has made significant progress (Li et al., 2024; Haas et al., 2024; Dou et al., 2024; Sarkar et al., 2024; Astruc et al., 2024; Dufour et al., 2025). Methodologically, GeoCLIP (Vivanco Cepeda et al., 2023) leverages the CLIP architecture to separately model images and GPS coordinates, retrieving the GPS candidate closest to the image's representations through vector similarity matching. Img2Loc (Zhou et al., 2024) is the first to introduce LVLMs into image geolocalization, retrieving similar images' information and incorporating it as prompts into the LVLM input to utilize the world knowledge acquired during pretraining to predict the image's location. G3 (Jia et al., 2024) further improves upon Img2Loc by optimizing both the image retrieval and reasoning processes, enabling the model to obtain more accurate reference information and fully exploit the prediction potential of LVLMs. GLOBE (Li et al., 2025a) enhances the reasoning ability of LVLMs through reinforcement learning, enabling backbone LVLMs can accurately infer the shooting location from images.

**Benchmark of Geolocalization.** Common evaluation datasets used in geolocalization tasks include IM2GPS (Hays & Efros, 2008) and YFCC (Thomee et al., 2016). On the benchmarking side, LLMGeo (Wang et al., 2024) collects datasets from Google Street View and evaluates models including GPT-4V (Achiam et al., 2023), Google Gemini (Team et al., 2023), BLIP (Li et al., 2023), Fuyu (Bavishi et al., 2023), InternLM-VL (Dong et al., 2024), and LLaVA (Liu et al., 2023). Liu et al. (2024) evaluates the performance of LVLMs on IM2GPS and YFCC and shows that incorporating Chain-of-Thought (CoT) (Wei et al., 2022) reasoning can improve performance on geolocalization tasks. Jay et al. (2025) also extracts data from Google Street View to create a more generalized evaluation set, finding that LVLMs already outperform the average human baseline in geolocalization capabilities. FairLocator (Huang et al., 2025) evaluates LVLMs' urban geolocalization abilities and focuses on biases in the geolocalization capabilities of LVLMs. In contrast to these methods and benchmark studies, we propose GeoArena, the *first dynamic and user-preference-based benchmark for image geolocalization*. GeoArena collects in-the-wild images uploaded by users, effectively mitigating the data leakage issues of static datasets while also avoiding the privacy risks associated with requiring exact GPS annotations. In addition, the image distribution in GeoArena is more representative of real-world use cases, and the platform uses user preferences to generate rankings of model capabilities. This provides a more robust and user-aligned evaluation framework for image worldwide geolocalization.

## 3 GEOARENA

GeoArena is an interactive platform designed to evaluate the geolocalization capabilities of various LVLMs. In this section, we provide a detailed description of GeoArena, including its live interface (Section 3.1), data collection process (Section 3.2), the models it encompasses (Section 3.3), and the ranking computation methods (Section 3.4).

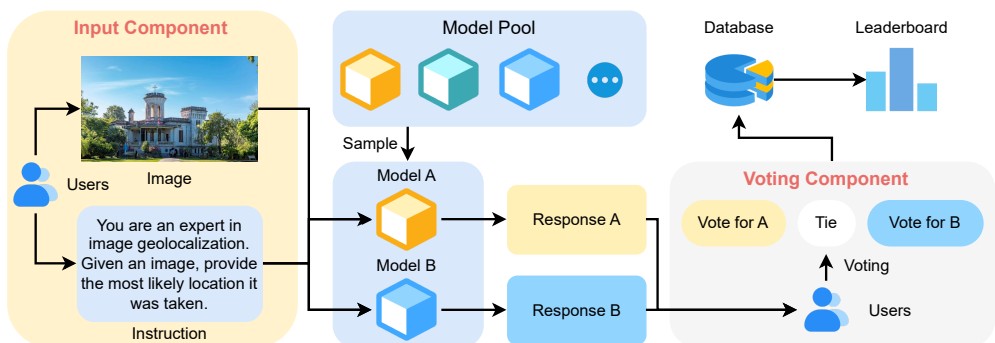

Figure 2: Overview of GeoArena.

### 3.1 LIVE INTERFACE

To facilitate user interaction, GeoArena is an online platform that allows any user to conveniently access the leaderboard and participate in data collection through a public link. As shown in Figure 2, the live interface consists of two main components: an input component and a voting component. (1) The input component includes both an image input and a prompt input. Users can upload images that they wish to geolocate, while the prompt input allows users to specify personalized geolocalization instructions. To improve efficiency, we also provide a default instruction derived from previous work (Zhou et al., 2024; Jia et al., 2024). (2) The voting component displays two side-by-side outputs generated by two anonymized models that are automatically sampled. After clicking the submit button, three voting options pop up: "vote for left", "vote for right", and "tie" (indicating comparable quality between the two outputs). Once the user submits a vote, the true model identities are revealed to maintain user impartiality during the voting process.

### 3.2 DATA COLLECTION

GeoArena collects essential data for each evaluation session to enable rigorous analysis and reliable leaderboard computation. For every voting event, we record the names of the two models being compared, the winning model, the user-provided prompt, the uploaded image, and the generated responses. This information ensures the traceability of each comparison, supports the calculation of rankings, and allows for reproducible experiments. All data are stored in structured JSON files, which facilitate downstream analysis and leaderboard generation. To preserve user privacy, we anonymize user inputs and apply filters to remove any potentially sensitive or inappropriate content.

### 3.3 PARTICIPATING MODELS

To ensure comprehensive and meaningful comparisons, GeoArena includes a wide range of both open-source and proprietary models. Our selection covers popular LVLMs from multiple providers. For the GPT series (Achiam et al., 2023), we include GPT 4o, GPT 4o mini, GPT 4.1, GPT 4.1 mini, and GPT 4.1 nano. From the Gemini family (Team et al., 2023), we incorporate Gemini 2.5 pro and Gemini 2.5 flash. The Claude series includes Claude Opus 4 and Claude Sonnet 4. We also evaluate Llama 4 maverick and Llama 4 scout (Touvron et al., 2023), as well as Gemma 3 models (Team et al., 2025) in sizes of 27B, 12B, and 4B. Additionally, our platform features Qwen 2.5 VL models in sizes of 72B, 32B, 7B (Bai et al., 2025). As shown in Table 2, in total, GeoArena currently benchmarks 17 models. This broad coverage ensures that users and researchers can evaluate model performance across different architectures, training paradigms, and capabilities.

### 3.4 RANKING COMPUTATION METHODS

**Online Elo Ranking.** The Elo rating system is a widely used approach to estimate the relative strength of different models or players based on pairwise comparisons. Originally introduced for ranking chess players, it has been extended to various evaluation tasks in machine learning and

Table 2: Large-scale models benchmarked in *GeoArena*. Prices are USD / million tokens (input/output) and USD / thousand (image).

| Model | Company | Params | Openness | API Price (input / output / image) |
|-------|---------|--------|----------|-------------------------------------|
| GPT 4o | OpenAI | Unknown | Proprietary | $2.50 / $10.00 / $3.61 |
| GPT 4o mini | OpenAI | Unknown | Proprietary | $0.15 / $0.60 / $0.22 |
| GPT 4.1 | OpenAI | Unknown | Proprietary | $2.00 / $8.00 / - |
| GPT 4.1 mini | OpenAI | Unknown | Proprietary | $0.40 / $1.60 / - |
| GPT 4.1 nano | OpenAI | Unknown | Proprietary | $0.10 / $0.40 / - |
| Gemini 2.5 flash | Google DeepMind | Unknown | Proprietary | $0.15 / $0.60 / $0.62 |
| Gemini 2.5 pro | Google DeepMind | Unknown | Proprietary | $1.25 / $10.00 / $5.16 |
| Claude Sonnet 4 | Anthropic | Unknown | Proprietary | $3.00 / $15.00 / $4.80 |
| Claude Opus 4 | Anthropic | Unknown | Proprietary | $15.00 / $75.00 / $24.00 |
| Llama 4 maverick | Meta | 17B/402B | Open-source | $0.15 / $0.60 / $0.67 |
| Llama 4 scout | Meta | 17B/109B | Open-source | $0.08 / $0.30 / - |
| Gemma 3 27B | Google | 27B | Open-source | $0.10 / $0.20 / $0.03 |
| Gemma 3 12B | Google | 12B | Open-source | $0.15 / $0.10 / - |
| Gemma 3 4B | Google | 4B | Open-source | $0.02 / $0.04 / - |
| Qwen 2.5 VL 72B | Alibaba | 72B | Open-source | $0.25 / $0.75 / - |
| Qwen 2.5 VL 32B | Alibaba | 32B | Open-source | $0.90 / $0.90 / - |
| Qwen 2.5 VL 7B | Alibaba | 7B | Open-source | $0.20 / $0.20 / - |

artificial intelligence. Elo rating provides an interpretable score that reflects the expected probability of one model outperforming another. Formally, given two models $i$ and $j$ with Elo ratings $R_i$ and $R_j$, the expected probability that model $i$ will outperform model $j$ is defined as:

$$E(i, j) = \frac{1}{1 + 10^{(R_j - R_i)/\alpha}} \tag{1}$$

where $\alpha$ is a scaling parameter that controls the spread of the probability function, typically set to 400 in most implementations. After observing the actual outcome $S(i, j)$, where $S(i, j) = 1$ if model $i$ wins, $S(i, j) = 0.5$ for a tie, and $S(i, j) = 0$ if model $i$ loses, the Elo rating of model $i$ will be updated as: $R'_i = R_i + K \cdot (S(i, j) - E(i, j))$, where $K$ is a learning rate that determines how quickly the rating adapts to new results. From the above description, we can summarize two key features of the Elo rating system. First, it can operate without requiring a complete history of past matches, updating each model's rating using only its current Elo rating and the outcome of its most recent match. Second, the Elo rating system inherently assumes that the strength of each participant changes over time, rather than remaining constant. However, in the context of evaluating LVLMs, we generally assume that model capabilities are static. Furthermore, Elo ratings are more strongly influenced by recent matches, making them highly sensitive to the order of matches—an effect that is undesirable in our setting. To address this, we follow prior work (Chiang et al., 2024) and apply the Bradley-Terry model (Bradley & Terry, 1952) to estimate the final Elo ratings for each model on the image geolocalization task, ensuring a stable and order-invariant ranking.

**Bradley-Terry Model.** The Bradley-Terry (BT) model provides a principled way to estimate the relative strength of competing models through pairwise comparisons. In this framework, each model $i$ is assigned a latent strength parameter $R_i$. The probability that model $i$ outperforms model $j$ is given by:

$$P(i > j) = \frac{1}{1 + 10^{(R_j - R_i)/\alpha}}, \tag{2}$$

where $\alpha$ is a scaling parameter (typically set to 400) that controls the spread of probabilities. The BT model estimates the parameters $R_i$ by maximizing the likelihood of all observed pairwise outcomes, accounting for repeated comparisons through a weighting term $W_{ij}$. The likelihood function is defined as:

$$\mathcal{L}(\mathbf{R}) = \sum_{i,j \in N, i \neq j} W_{ij} \log \left( \frac{1}{1 + 10^{(R_j - R_i)/\alpha}} \right) \tag{3}$$

To compute the final Elo ratings, we apply a linear transformation to align the model scores with the Elo rating scale. Specifically, after fitting the BT model via logistic regression, the estimated

Table 3: GeoArena Leaderboard in September 2025.

| Ranking | Model | ELO Rating | 95% CI lower | 95% CI upper |
|---|---|---|---|---|
| 0 | Gemini 2.5 pro | 1319.7 | 974.8 | 1443.8 |
| 1 | Gemini 2.5 flash | 1206.5 | 1062.2 | 1330.6 |
| 2 | Qwen 2.5 VL 72B | 1094.5 | 982.6 | 1181.9 |
| 3 | Gemma 3 12B | 1086.5 | 1002.6 | 1186.4 |
| 4 | Gemma 3 27B | 1065.5 | 959.3 | 1159.8 |
| 5 | GPT 4.1 mini | 1059.8 | 970.0 | 1161.4 |
| 6 | Llama 4 maverick | 1046.6 | 944.6 | 1115.3 |
| 7 | Qwen 2.5 VL 32B | 1044.8 | 964.9 | 1119.0 |
| 8 | GPT 4.1 | 1044.8 | 964.9 | 1119.0 |
| 9 | Claude Opus 4 | 1042.3 | 933.8 | 1130.0 |
| 10 | Gemma 3 4B | 1027.3 | 936.3 | 1102.0 |
| 11 | Claude Sonnet 4 | 1019.9 | 921.3 | 1113.8 |
| 12 | GPT 4o | 1000.0 | 1000.0 | 1000.0 |
| 13 | Llama 4 scout | 984.2 | 876.0 | 1077.1 |
| 14 | Qwen 2.5 VL 7B | 950.9 | 868.4 | 1056.2 |
| 15 | GPT 4.1 nano | 917.9 | 819.1 | 1015.5 |
| 16 | GPT 4o mini | 871.6 | 715.2 | 1114.7 |

parameters $\hat{R}_i$ are transformed as: $\text{Elo}_i = \text{scale} \cdot \hat{R}_i + \text{init\_rating}$, where scale is typically set to 400 and init_rating is set to 1000. This transformation preserves the relative ranking among models while making the scores more interpretable and consistent with standard Elo rating conventions.

**Confidence Interval.** To ensure that the model ranking results are not overly dependent on a particular sample of comparisons, we estimate confidence intervals (CIs) for the elo scores. Specifically, we adopt a bootstrap procedure similar to the methodology employed in Chatbot Arena (Chiang et al., 2024), which repeatedly resamples the battle outcomes and re-computes the rating estimates. This approach allows us to quantify the variability in model rankings and provides statistically grounded intervals around each estimate. The inclusion of confidence intervals is essential because it enables us to distinguish between meaningful performance differences and those that may arise due to sampling noise. As a result, our reported rankings are not only more robust but also more interpretable from a statistical perspective, offering stronger evidence of the relative strengths of different LVLMs on the image geolocalization task.

## 3.5 GEOARENA-1K DATASET

Based on GeoArena, we further release the **GeoArena-1K** dataset. This dataset consists of samples each containing the user-uploaded image, the textual instructions, pairwise model responses, the names of the competing models, and the corresponding user voting outcomes. To the best of our knowledge, this is the first user preference dataset for LVLMs in the domain of image geolocalization. Beyond serving as a preference dataset on image geolocalization, GeoArena-1K provides a valuable resource for advancing research in reward modeling and the development of geographic foundation models. More details about the GeoArena-1K dataset are illustrated in Appendix A.1.

## 4 BENCHMARKS AND RESULTS ANALYSIS

### 4.1 ARENA LEADERBOARD

Table 3 presents the GeoArena leaderboard as of September 2025. The reported 95% confidence intervals (CI lower, CI upper) are computed via bootstrap resampling over 100 rounds, capturing rating variability under different voting subsets. To ensure the data aligns with the geolocalization task, we manually filtered the user-uploaded prompts. From this table, several key observations can be drawn: (1) Gemini models from DeepMind achieve the strongest performance, with Gemini 2.5 pro (Elo 1319.7) and Gemini 2.5 flash (Elo 1206.5) clearly outperforming all other systems. This

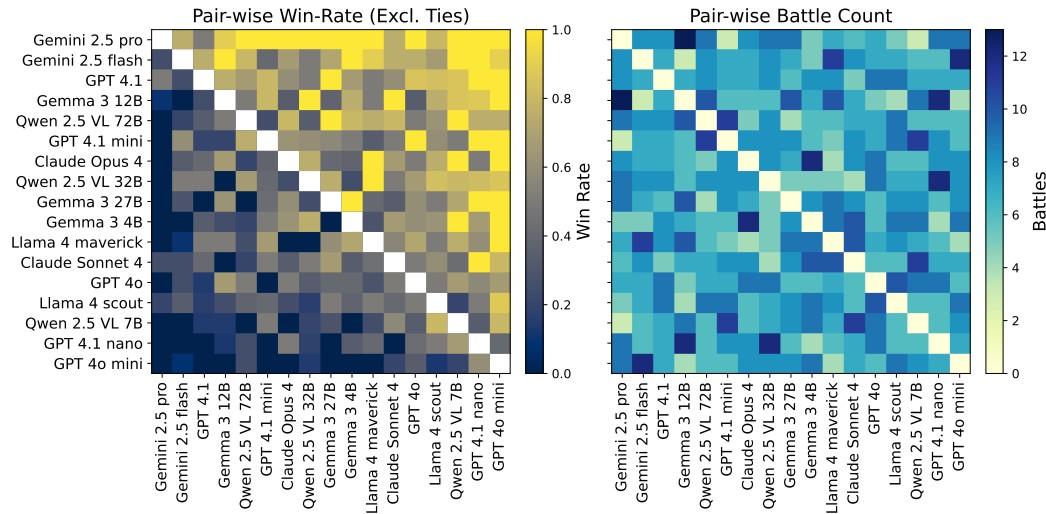

Figure 3: Pair-wise Performance Comparison of Models (Win-Rate and Battle Count).

highlights the advantage of large-scale, production-level multimodal pre-training in the challenging image geolocalization task. (2) Open-source families such as Qwen 2.5 and Gemma 3 obtain competitive rankings. For example, Qwen 2.5 VL 72B (Elo 1094.5) surpasses Gemma 3 12B (Elo 1086.5) and performs comparably to the GPT 4.1 series, suggesting that open-source initiatives are rapidly narrowing the gap with proprietary frontier systems. (3) Several models, including Llama 4 maverick, GPT 4.1, and Claude Opus 4, cluster within the Elo 1040-1050 range. Their confidence intervals overlap substantially, indicating no statistically significant differences between these models. (4) Smaller variants such as GPT 4.1 nano, GPT 4o mini, and the lightweight Qwen model (Qwen 2.5 VL 7B) exhibit clear performance degradation, with ratings below 960. This underscores the inherent difficulty of image geolocalization, where reduced model capacity limits generalization across diverse global contexts. (5) The wide rating spread (1320 down to 870) validates the discriminative power of GeoArena. It provides a reliable platform to distinguish frontier-level systems from lightweight baselines, which is crucial for advancing research on geospatial reasoning in LVLMs.

## 4.2 BATTLE DATA ANALYSIS

To provide a comprehensive view of comparative model performance, we conduct a pair-wise analysis of model battles, reporting both win-rates and battle counts. Figure 3 reports a pair-wise comparison across models, with the left panel showing head-to-head win rates (ties excluded) and the right panel showing the corresponding battle counts. Models are ordered by their average win rate, which makes the block structure of the heatmap interpretable. We can find: **(1) Frontier models consistently dominate.** Gemini 2.5 pro, Gemini 2.5 flash, and GPT 4.1 occupy the top rows, maintaining win rates close to or above 0.7 against nearly all competitors. Their advantage is not limited to small baselines but extends to strong models from other families, suggesting that both model capacity and advanced alignment procedures contribute to their robustness. **(2) Mid-scale systems show transitional behavior.** Models such as Gemma 3 12B, Qwen 2.5 VL 72B, and GPT 4.1 mini occupy the middle tier. They achieve favorable outcomes against smaller instruction-tuned variants but exhibit substantial performance gaps when challenged by the frontier tier. This indicates a stepwise stratification that correlates with effective model size and tuning intensity. **(3) Lower-capacity models underperform broadly.** Systems including Gemma 3 4B, Qwen 2.5 VL 7B, GPT 4.1 nano, and GPT 4o mini cluster near the bottom of the heatmap, with win rates typically below 0.3 against larger peers. Their deficits are systematic across families, reflecting limited parameter budgets and less extensive post-training data. **(4) Family-specific patterns emerge.** Within families, performance scales predictably with size. For example, the Qwen 2.5 VL series shows clear gains moving from 7B to 72B parameters. These intra-family trends suggest that scaling and alignment strategies jointly determine competitiveness.

Table 4: Agreement Analysis between Expert and Crowd.

| Expert \ Crowd | Left Win | Tie | Right Win | Agreement Rate |
|---|---|---|---|---|
| Left Win | 30 | 3 | 3 | 83.3% |
| Tie | 5 | 21 | 6 | 65.6% |
| Right Win | 2 | 3 | 27 | 84.4% |
| Agreement Rate | 81.1% | 77.8% | 75.0% | 78.0% |

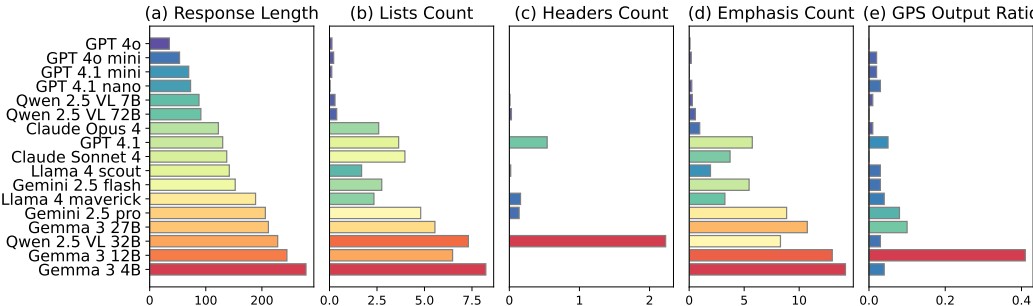

Figure 4: Distribution of Style Features in Model Outputs.

## 4.3 RELIABILITY ANALYSIS OF VOTING

To validate the quality of the voting data, we randomly sample 100 examples from the dataset and have expert annotators evaluate them. Specifically, given an image to be geolocalized, a textual prompt, and two anonymized model responses, the expert is asked to judge which response is better, or to select a tie if applicable. Expert is allowed to use any external tools, including search engines, to assist their decision-making. On average, each evaluation takes approximately 3–5 minutes. Table 4 presents the distribution of preferences between experts and crowds on the sampled examples. Overall, we observe a consistently high agreement rate between expert and crowd annotations, typically ranging from 75% to 85%, with an average agreement of 78%. *According to prior studies (Chiang et al., 2024), this constitutes a strong agreement level, supporting the reliability of the collected voting data.*

## 4.4 PREFERENCE ANALYSIS

To better understand which characteristics of model responses drive user preference, following previous work (Chiang et al., 2024; Tianle Li, 2024; Dubois et al., 2024), we extend the standard Bradley-Terry regression framework by incorporating style-related features as confounding variables. In practice, this means that for each model comparison, we not only encode which two models are being compared, but also include additional features that describe the differences in response style, such as normalized response length, number of lists, headers, emphasis markers, or the ratio of GPS outputs. By including these features in the regression, we can separate the effect of style from the intrinsic ability of the model. The style coefficients ($\beta$) are estimated via logistic regression, where style features are normalized and included alongside model indicators in an extended Bradley-Terry design matrix. The style coefficients, therefore, quantify how much specific stylistic traits influence user choices. A higher coefficient $\beta$ for a style feature indicates that this attribute contributes more positively to user preference. In this study, we primarily consider five different style features: response length (measured by the number of words), lists count (including both unordered and ordered lists), headers count, emphasis count (including the number of bold and italic items), and GPS output ratio

Table 5: Estimated Influence of Style Features on User Preference. A higher coefficient ($\beta$) for a style feature indicates that this attribute contributes more positively to user preference.

| Features | Response Length | Lists Count | Headers Count | Emphasis Count | GPS Output Ratio |
|---|---|---|---|---|---|
| Coefficient $\beta$ | 0.526 | 0.095 | -0.153 | -0.117 | 0.06 |

(the proportion of responses containing GPS-level predictions). Figure 4 illustrates the distribution of these features across different models, showing clear stylistic variation in model outputs. From the experimental results in Table 5, we observe consistent findings with prior work (Chiang et al., 2024; Steyvers et al., 2024; Tianle Li, 2024): response length exhibits a strong positive correlation with human preference (i.e., $\beta_{\text{response}} = 0.526$), as longer responses are more likely to be favored by users. In addition, both lists count ($\beta_{\text{list}} = 0.095$) and GPS output ratio ($\beta_{\text{GPS}} = 0.06$) are positively correlated with preference, where a higher number of lists often reflects more explicit reasoning steps, and GPS outputs provide finer-grained and more concrete answers. However, headers count and emphasis count do not show positive associations with human preference. A possible explanation is that excessive use of structural markers or textual emphasis may be perceived as superficial formatting rather than substantive content, and thus does not contribute to the perceived quality or informativeness of the response.

## 4.5 ALIGNMENT STUDY BETWEEN LVLM AND USER

To further examine whether LVLMs can serve as reliable judges for geolocalization responses, we conduct an alignment study that compares LLM preferences with human annotations. Specifically, we randomly sample 100 response pairs from the dataset. For each pair, we ask an LVLM to decide which response (generated by Model A or Model B) is better in terms of accuracy, reasoning, and clarity, and require the model to output

Table 6: Alignment accuracy of LLMs with human judgments on sampled response pairs.

| Model | Accuracy |
|---|---|
| Gemini 2.5 pro | 0.6579 |
| Qwen 2.5 VL 72B | 0.4667 |

only one of three labels: win, tie, or loss. The prompt template we used is shown in Appendix A.2. We then compute the agreement accuracy between the LVLM's judgment and the human-provided ground truth labels. Table 6 reports the alignment results for two representative models, the top-performing proprietary and open-source models on the GeoArena leaderboard, Gemini 2.5 pro and Qwen 2.5 VL 72B. The results show that Gemini 2.5 pro achieves a substantially higher agreement rate (65.79%) with human evaluations compared to Qwen 2.5 VL 72B (46.67%). This suggests that Gemini 2.5 pro exhibits stronger alignment with human judgment in assessing geolocalization task responses. These findings highlight that while LLMs can approximate human preferences to a certain extent, significant gaps remain. This motivates future work on designing more faithful and robust LLM-based evaluators for geolocalization and other multimodal tasks.

## 4.6 CASE STUDY

To illustrate our framework, we present a case study using an image of the Ngātoroirangi Māori Rock Carvings at Mine Bay on Lake Taupō, New Zealand. As shown in Figure 1, different models exhibit varying levels of reasoning depth and factual accuracy. Gemini 2.5 pro produces a comprehensive analysis, identifying salient visual features such as the Māori face carving, surrounding cliffs, and water-based accessibility, while also providing historical and cultural context (e.g., the carving's creation in 1980 by Matahi Whakataka-Brightwell). In contrast, GPT 4o mini generates only a brief description, lacking explicit reasoning and omitting cultural details. This comparison underscores the importance of reasoning quality and contextual grounding in geolocalization tasks, showing that structured analyses align more closely with human preferences and task requirements. We also give hard cases analysis and more case studies in Appendix A.3 and Appendix A.4.

## 5 CONCLUSION

In this work, we present GeoArena, a dynamic and user-preference-based benchmarking platform for evaluating LVLMs on worldwide image geolocalization tasks. By collecting in-the-wild user-submitted images and integrating pairwise user preference evaluations, GeoArena overcomes the limitations of existing static benchmarks that often suffer from data leakage, insufficient reasoning assessment, and privacy issues. We implement a stable Bradley-Terry model, enabling reliable and interpretable ranking of models under diverse real-world conditions. Overall, GeoArena offers a practical, scalable, and user-aligned framework that bridges the gap between automated metrics and human evaluation. We believe GeoArena will facilitate future research in LVLMs and GeoAI, providing valuable resources for developing robust, generalizable, and user-preference-aligned geolocalization systems.

ETHICS STATEMENT

Our work involves the development and deployment of GeoArena, an open benchmarking platform that evaluates LVLMs on worldwide image geolocalization tasks using real-world images and human preferences. Our ethical statement is detailed as follows:

1. We prioritize user privacy and data protection. GeoArena does not collect or store any personally identifiable information, and users are not required to submit GPS coordinates or metadata tied to private locations. All uploaded images and preference votes are anonymized and stored in compliance with ethical data management practices.

2. Our human evaluation is limited to pairwise preference voting and does not involve sensitive demographic or personal data. No compensation or recruitment was involved, and the voting interface includes disclaimers and consent mechanisms.

3. Our benchmark is explicitly designed for research purposes, with all model outputs and analysis made publicly available to support transparent and responsible evaluation. The voting data will be released under appropriate open data licenses for research use only.

We confirm that this work complies with the ICLR Code of Ethics.

**Disclosure of LLM Usage**    We only used Large Language Models (LLMs) to aid or polish the writing in this work.

REPRODUCIBILITY STATEMENT

To promote reproducibility and further research, we will publicly release both the source code and the collected data of GeoArena. This includes the full platform backend and frontend codebase (for image upload, voting, and model evaluation), as well as all anonymized user-submitted images, prompts, and pairwise voting records. We aim to support the community in building similar arena-style benchmarks for other tasks. The repository will include documentation and deployment instructions to facilitate reuse and adaptation across domains.

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

# A   APPENDIX

## A.1   DATASET CHARACTERISTICS AND COMPOSITION

Table 7: Composition of Image Features in GeoArena-1K Dataset.

| Attribute | Category | Percentage |
|---|---|---|
| Scene Type | Outdoor | 94.2% |
| | Indoor | 5.8% |
| Text Presence | Has Text | 45.2% |
| | No Text | 54.8% |
| Landmark Presence | Has Landmark | 15.8% |
| | No Landmark | 84.2% |

To further explore the characteristics of the GeoArena-1K dataset, we employ GPT 4o to annotate the collected images, focusing on three key aspects:

1. Scene Type: whether the image depicts an indoor or outdoor setting.

2. Text Presence: whether the image contains prominent, recognizable text.

3. Landmark Presence: whether the image features a landmark, such as a historical site or natural icon.

The corresponding results are presented in Table 7. The figure comprises three doughnut charts, each illustrating the distribution of one of the annotated attributes across the GeoArena-1K dataset: Indoor/Outdoor Distribution:

1. The first doughnut chart indicates that 94.2% of images are classified as outdoor scenes, with only 5.8% representing indoor environments. This pronounced skew toward outdoor imagery aligns with the global scope of GeoArena, where user-submitted images are likely dominated by exterior scenes captured in diverse geographic contexts.

2. Text Presence: The second doughnut chart reveals a more balanced distribution, with 54.8% of images lacking recognizable text ("no text") and 45.2% containing text ("has text"). This near-equitable split underscores the dataset's richness, incorporating both text-free natural scenes and images with textual elements such as signs or labels. This variability is particularly valuable for assessing LVLM capabilities in multi-modal reasoning, where text recognition can enhance location prediction accuracy.

3. Landmark Presence: The third doughnut chart shows that 84.2% of images do not contain landmarks ("no landmark"), while 15.8% do ("has landmark"). The low prevalence of landmarks reflects the dataset's emphasis on general geographic scenes rather than iconic or tourist-heavy locations, offering a broad representation of natural and urban environments worldwide. This distribution highlights GeoArena-1K's potential to test LVLM generalization across less distinctive locales, a challenging yet realistic scenario for global geolocalization. Overall, these distributions reveal the GeoArena-1K dataset's heterogeneity, making it a robust resource for benchmarking LVLM performance under real-world conditions.

## A.2   LVLM ALIGNMENT EVALUATION PROMPT

The prompt template used for LVLM alignment evaluation is as follows:

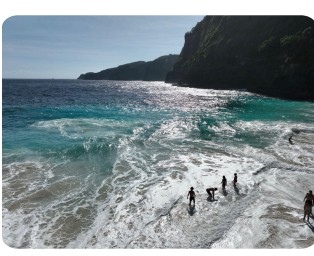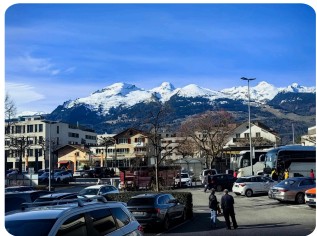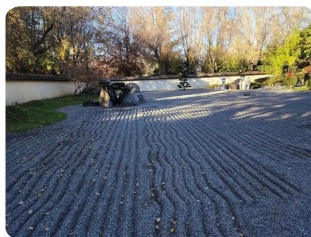
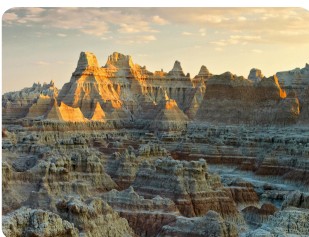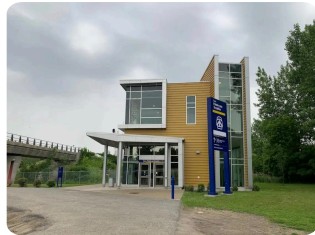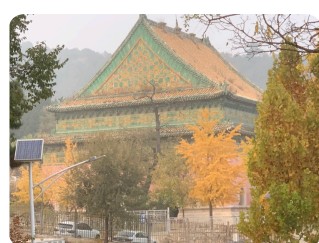

Figure 5: Case Study: Images Where Strong Models Excel but Weaker Models Fail

---

You are an expert evaluator in image geolocation tasks. I will give you two model responses
to the same geolocation prompt.
**Here is the prompt:**
- Prompt: `{sample['prompt']}`
- Image: `{sample['image']}`
**Response A:** `{sample['response A']}`
**Response B:** `{sample['response B']}`
**Your task is to decide which response is better based on:**
1. Accuracy of the predicted location
2. Strength of reasoning and evidence
3. Clarity and specificity
**Output only one word:**
- "win" if Response A is better
- "loss" if Response B is better
- "tie" if both are equally good

---

We first prompt the LVLM to act as an expert evaluator for the geolocalization task. For each sample,
the model is provided with the prompt and the associated images, along with the responses from
two candidate models. It is then asked to determine which response is better, considering three
dimensions: accuracy, reasoning, and clarity and specificity.

## A.3 HARD CASES ANALYSIS

To better understand the distinction between top-tier and lower-tier models, we examined a set of
representative cases where strong models predicted correctly but weaker models failed, as illustrated
in the Figure 5. Specifically, we selected the top-5 models on the leaderboard—Gemini 2.5 pro,
Gemini 2.5 flash, Qwen 2.5 VL 72B, Gemma 3 12B, and Gemma 3 27B—as well as the bottom-5
models—GPT 4o, Llama 4 scout, Qwen 2.5 VL 7B, GPT 4.1 nano, and GPT 4o mini. We then
extracted examples where the top-5 models outperformed the bottom-5 models. These images
generally do not contain prominent or unique landmarks that can be easily recognized. Instead, they
often depict environments that require reasoning over subtle contextual cues–such as vegetation
types, architectural style, road structures, or geographic textures–to infer the location. This suggests
that strong models may better leverage fine-grained reasoning and background knowledge beyond
surface-level features. The presence of such challenging samples highlights a promising direction
for future model development: expanding training and evaluation on these harder cases. By doing

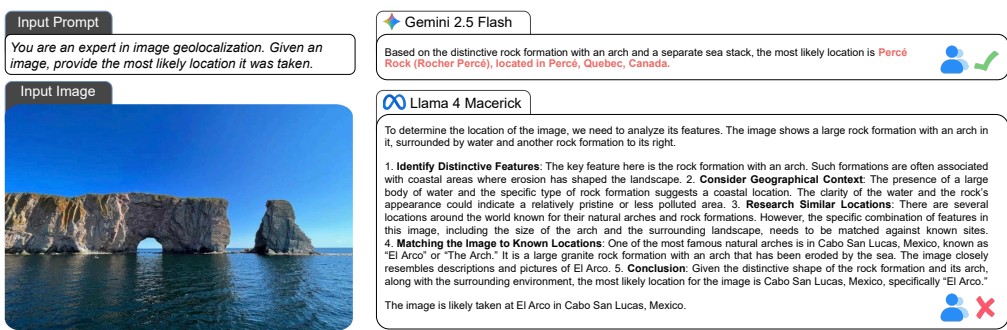

Figure 6: Additional Case Study: Identifying the Percé Rock.

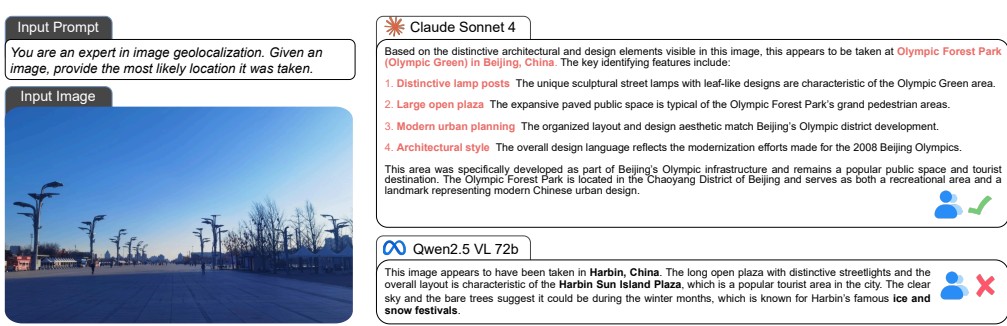

Figure 7: Additional Case Study: Identifying the Olympic Park, Beijing.

so, upcoming systems may gain robustness and achieve more consistent performance across diverse scenarios, particularly where visual clues are subtle and non-obvious.

### A.4 ADDITIONAL CASES FROM GEOARENA

In this section, we present additional case studies from GeoArena to illustrate the diversity and complexity of the geolocalization tasks it encompasses. Figure 6 showcases an image of the iconic Percé Rock in Quebec, Canada, highlighting the model's ability to recognize unique geological formations. Figure 7 features the Olympic Park in Beijing, China, demonstrating the model's proficiency in identifying modern architectural landmarks. Lastly, Figure 8 depicts a golf course in Fiji, emphasizing the model's capability to infer locations based on environmental and recreational context. These examples underscore GeoArena's effectiveness in challenging models to perform accurate geolocalization across a wide range of scenarios.

### A.5 VERIFYING THE FEASIBILITY OF AUTOMATIC PROMPT FILTERING

To maintain a reliable leaderboard, it is essential to ensure that user inputs are relevant to the image's geolocation. Although we currently remove inappropriate or sensitive queries manually, this approach may not scale as the platform expands. To examine whether LLMs can replace manual filtering, we conduct an experiment to determine whether LLMs can identify when a user prompt requests geolocating an image.

For this study, we construct a binary classification task. We randomly select 100 prompts from our voting data and assign them the label True, indicating that they ask about image geolocation. In parallel, we sample 100 prompts from the Chatbot Arena dataset[2], which contains general-purpose prompts, and labeled them as False. Each model is given a simple instruction that defines image

---

[2] https://huggingface.co/datasets/lmsys/chatbot_arena_conversations

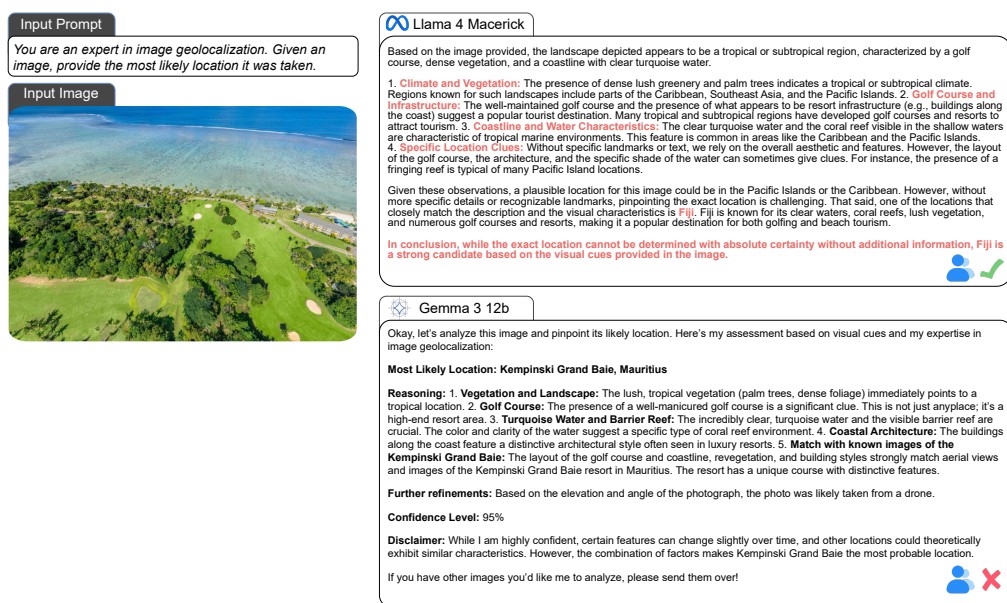

Figure 8: Additional Case Study: Identifying the Golf Course in Fiji.

geolocation, specifies the expected JSON output, and directs the model to respond only with a True or False label. The instruction is given as follows:

> You are a prompt classifier. Analyze the provided user prompt and determine if it is asking about image geolocalization.
> Image geolocalization refers to determining or estimating the geographic location (e.g., city, country, landmark) where an image was taken based on its visual content.
> Return ONLY a JSON object with one key: "is_geo". The value must be "true" if the prompt is inquiring about geolocalizing an image (e.g., "Where was this photo taken?" or instructions for an expert in image geolocalization), or "false" otherwise. If uncertain, default to "false".
> Output format (no extra words): "is_geo": "true"|"false"
> User prompt: user_prompt

We evaluate three models: Gemini 2.0 flash, GPT 3.5 turbo, and GPT 4.1 mini. All three models achieve 100% accuracy on this task. The high accuracy is mainly due to two factors. First, most users ask questions through the default prompt provided by GeoArena, which reflects a stable phrasing pattern. Second, prompts that request geolocalization usually contain explicit references to places, images, or location inference, which makes them easy for the models to detect. These observations show that modern language models can serve as reliable automatic filters for user inputs. Such a mechanism would allow the leaderboard to remain focused on geolocalization queries while reducing the need for manual inspection.

## A.6  USER CONSENT

To ensure responsible data usage and protect user privacy, GeoArena requires all participants to provide consent before submitting any images or preference votes. When users interact with the platform, they are presented with a clear consent statement indicating that uploaded images and voting records may be used for research purposes and may be released in anonymized form. Users are also informed that participation is voluntary and that they should avoid uploading sensitive or personally identifiable content. These measures confirm that the data included in GeoArena is collected with explicit user permission and used strictly within an academic context.

