# OpenReview forum: "GeoArena: An Open Platform for Benchmarking Large Vision-language Models on WorldWide Image Geolocalization"
_ICLR.cc/2026/Conference — ICLR 2026 Conference Withdrawn Submission_

### Official Review · Reviewer_zWjz · 2025-10-27

**Soundness:** 3
**Presentation:** 3
**Contribution:** 2
**Rating:** 6
**Confidence:** 5

**Summary:**

This paper presents a benchmarking platform, called GeoArena, for evaluating LVLMs on global image geolocalization tasks. This platform was deployed online for three months, and collected thousands of voting records for the subsequent ranking and detailed analysis. Notably, GeoArena stands from a new perspective, which differs from the existing benchmarks, using the user-upload images and user-preference-based evaluation to address the issues of data leakage and privacy.

**Strengths:**

- This paper is well-written and easy to follow
- The perspective presented in this work is novel and interesting
- The detailed analyses are insightful and valuable

**Weaknesses:**

Although I personally find the perspective of this paper quite compelling, I would like to raise a few concerns and look forward to the authors’ responses:
- The design of user-uploaded images is motivated by the potential data leakage in pretraining datasets. However, how do you ensure that a user-uploaded image was actually taken by the user themselves, rather than selected from online sources? Is there any verification or filtering mechanism in place to prevent such cases?
- The paper states that GeoArena collected thousands of voting records over three months, but does not specify the number of participating users. Could the authors provide more detailed statistics about the voting data, particularly the number of unique users? If a single user contributed the majority of the uploaded images and corresponding votes, this could introduce significant bias into the collected data
- This benchmark is built on user preference and does not include prediction accuracy. I still have concerns about voting (the sole evaluation method), because the task of geolocalization has a ground truth. If we neglect this important metric, what is the significance of performing geolocalization —merely to describe the scene? I would like to hear the authors’ discussion on this point, as it is directly related to the motivation of the paper
- In Table 3, the ranking results show that GPT‑4.1 mini outperforms GPT‑4.1, which appears inconsistent with the official API documentation describing GPT‑4.1 as more capable. Could the authors provide an explanation for this observation?

**Questions:**

Given some remaining concerns, I am currently giving a borderline accept score. I would like to see the authors’ rebuttal before deciding whether to raise my score.

---

> ### Author Response · Authors · 2025-11-20
> **Response [1/2]**
>
> > The design of user-uploaded images is motivated by the potential data leakage in pretraining datasets. However, how do you ensure that a user-uploaded image was actually taken by the user themselves, rather than selected from online sources? Is there any verification or filtering mechanism in place to prevent such cases?
>
> Thank you for raising this concern. We acknowledge that it is not feasible to guarantee that every uploaded image was personally taken by the user. Our goal is not to enforce authorship verification, but to reduce the systematic data leakage that occurs when evaluation sets are drawn directly from large public datasets (e.g., YFCC4K [1] and IM2GPS series [2]) that LVLMs were possibly pretrained on. User-uploaded images, even if occasionally selected from the internet, are far less likely to overlap with pretraining corpora in a structured or large-scale manner.
>
> In practice, we manually filter inputs that are not aligned with the geolocalization objective, as well as those that are obviously duplicated from public datasets, which is feasible at our current scale.
>
> Inspired by your comment, we conducted an additional experiment using LLMs as classifiers to determine whether a given prompt is indeed a geolocalization-related query.
>
> We sampled 100 prompts from our voting data and labeled them as True (indicating they inquire about geolocating an image). We also sampled 100 prompts from the Chatbot Arena dataset [3] and labeled them as False. We then asked several LLMs (Gemini 2.0 Flash, GPT-3.5-Turbo, and GPT-4.1-Mini) to classify each prompt using the instruction:
>
> ```
> You are a prompt classifier. Analyze the provided user prompt and determine if it is asking about image geolocalization.
>
> Image geolocalization refers to determining or estimating the geographic location (e.g., city, country, landmark) where an image was taken based on its visual content.
>
> Return ONLY a JSON object with one key: "is_geo".
> The value must be "true" if the prompt is inquiring about geolocalizing an image (e.g., "Where was this photo taken?" or instructions for an expert in image geolocalization), or "false" otherwise.
> If uncertain, default to "false".
>
> Output format (no extra words):
> {{"is_geo": "true"|"false"}}
>
> User prompt: {user_prompt}
> ```
>
> All three models achieved 100% accuracy. This is largely because most users adopt the default prompt provided by GeoArena, and prompts that request geolocation exhibit clear and recognizable linguistic patterns. These findings suggest that we can reliably leverage large language models to filter user inputs, ensuring that the leaderboard remains focused on geolocalization queries and therefore more trustworthy.
>
> > The paper states that GeoArena collected thousands of voting records over three months, but does not specify the number of participating users. Could the authors provide more detailed statistics about the voting data, particularly the number of unique users? If a single user contributed the majority of the uploaded images and corresponding votes, this could introduce significant bias into the collected data
>
> Thank you for the suggestion. Distinguishing unique users is challenging in our setting. Using access patterns would introduce large inaccuracies, as the same user may access the platform from different locations or devices. Requiring users to disclose identifiers would raise privacy concerns and could discourage participation. We appreciate the reviewer’s point and agree that understanding user diversity is valuable. We are open to adding such functionality if a privacy-preserving and reliable solution can be identified, and we will continue to explore possible approaches in future updates.
>
> > This benchmark is built on user preference and does not include prediction accuracy. I still have concerns about voting (the sole evaluation method), because the task of geolocalization has a ground truth. If we neglect this important metric, what is the significance of performing geolocalization —merely to describe the scene? I would like to hear the authors’ discussion on this point, as it is directly related to the motivation of the paper
>
> Thank you for raising this important point. We agree that accuracy is a fundamental metric in geolocalization, and our goal is not to replace accuracy-based evaluation. Instead, GeoArena provides a complementary perspective by measuring human-aligned reasoning quality, which accuracy alone cannot capture. To ensure that preference signals remain meaningful for geolocalization, we conducted an experiment in which expert annotators independently evaluated a subset of voting pairs (Section 4.3). According to prior studies [4], their judgments showed strong agreement with the crowd votes, indicating that user preference is closely aligned with actual geolocalization correctness rather than merely scene description. We have highlighted this validation more clearly in the revised version.

---

> > ### Author Response · Authors · 2025-11-20
> > **Response [2/2]**
> >
> > > In Table 3, the ranking results show that GPT‑4.1 mini outperforms GPT‑4.1, which appears inconsistent with the official API documentation describing GPT‑4.1 as more capable. Could the authors provide an explanation for this observation?
> >
> > Thank you for pointing this out. We agree that the observation may seem surprising given the general capability differences reported in the official API documentation. One possible explanation is that the training data or optimization focus of GPT-4.1 may differ from that of GPT-4.1 mini, and geolocalization is a relatively niche task that may not align with the primary objectives of the larger model. As a result, performance differences on this specific task may not fully correlate with overall model capability rankings. We believe such discrepancies are expected when evaluating models on specialized domains, and our findings highlight the value of task-specific benchmarking.
> >
> > [1] Thomee, Bart, et al. "Yfcc100m: The new data in multimedia research." Communications of the ACM 59.2 (2016): 64-73.
> >
> > [2] Hays, James, and Alexei A. Efros. "Im2gps: estimating geographic information from a single image." 2008 ieee conference on computer vision and pattern recognition. IEEE, 2008.
> >
> > [3] https://huggingface.co/datasets/lmsys/chatbot_arena_conversations
> >
> > [4] Chiang, Wei-Lin, et al. "Chatbot arena: An open platform for evaluating llms by human preference." *Forty-first International Conference on Machine Learning*. 2024.

---

> > > ### Comment · Reviewer_zWjz · 2025-11-24
> > >
> > > Thank you for the responses and I also read your rebuttals to the other reviewers.
> > > I strongly agree with the authors' viewpoints that "the reasoning process is an essential part of geolocalization" and "Pure distance-based scoring assigns these cases an extremely low score, which does not reflect their actual reasoning quality". I think this paper presents a very predictive benchmark with the context of MLLM reasoning in image geolocalization. Future benchmarks should also assess how well a model’s reasoning aligns with user preferences or step-wise quality judgments, beyond just its final geolocalization accuracy.
> > >
> > > Since the authors' reply has addressed most of my concerns, I will raise my score to support this work. I also hope the authors will further improve the paper based on the reviewers’ comments.

---

> > > > ### Author Response · Authors · 2025-11-26
> > > > **Response**
> > > >
> > > > Thank you for your positive feedback and for your support of this work. We are committed to further improving the paper based on the constructive comments provided.

---

### Official Review · Reviewer_ERsX · 2025-10-28

**Soundness:** 3
**Presentation:** 3
**Contribution:** 2
**Rating:** 4
**Confidence:** 4

**Summary:**

The paper presents GeoArena, a platform to perform blind pairwise comparisons between two VLMs randomly sorted from a pool of VLMs. The platform has been online for months, and has collected over a thousand votes. These votes, including the input images, prompts, and voting data, will be made publicly available

**Strengths:**

The idea is new. I am not aware of any similar existing platform where models can be evaluated solely on their geolocalization capabilities.
The data might be useful for certain tasks, although not clear for which tasks, and it is not clear if it can be released.

**Weaknesses:**

1) The platform does not measure geolocalization accuracy, but measures how likely a human is to choose a VLM's (potentially wrong) answer related to geolocalization. It is unclear how this can be useful. While evaluating VLMs on geolocalization accuracy is useful (there are many papers on this subject), and evaluating VLMs on human helpfulness is useful as well (which is what LMArena does), it is not clear how combining the two things can be useful, because it provides a mixed signal which might be uncorrelated to its geolocalization capability. Table 5 suggests that users prefer longer responses, with a coefficient β of 0.526, but there is no indication that higher scores (or longer responses) actually correspond to better geolocalization capabilities. To summarize, I don't see what conclusions I can draw by seeing the leaderboard: a better model in the leaderboard will be better at making the user think that it found the right place? How is this useful to the research community?

2) GeoArena-1K is composed of images uploaded by users and their preference response. Can this data be publicly released? Have the users been asked for consent? I could not find a link to GeoArena in the paper hence I couldn't try it myself to see if there is a button to consent to the release of my preference data and uploaded images. If not, that data can't be released.

3) The prompt is provided by the user, meaning that the user could easily ask anything about an image. Is there anything that ensures that the platform is constrained to geolocalization, or can the platform be used for any kind of image-text "battle" between two VLMs?
Line 049 says that a limitation of GPS-based evaluation is that existing evaluations only consider the spatial distance between the final prediction and the ground truth, ignoring the model’s reasoning process. Why is that a limitation? In most cases the user would only care about having an accurate geolocalization

**Questions:**

Questions included in weaknesses above

---

> ### Author Response · Authors · 2025-11-20
> **Response [1/2]**
>
> > The platform does not measure geolocalization accuracy, but measures how likely a human is to choose a VLM's (potentially wrong) answer related to geolocalization. It is unclear how this can be useful. While evaluating VLMs on geolocalization accuracy is useful (there are many papers on this subject), and evaluating VLMs on human helpfulness is useful as well (which is what LMArena does), it is not clear how combining the two things can be useful, because it provides a mixed signal which might be uncorrelated to its geolocalization capability. Table 5 suggests that users prefer longer responses, with a coefficient β of 0.526, but there is no indication that higher scores (or longer responses) actually correspond to better geolocalization capabilities. To summarize, I don't see what conclusions I can draw by seeing the leaderboard: a better model in the leaderboard will be better at making the user think that it found the right place? How is this useful to the research community?
>
> Thank you for this valuable question. Our goal is not to replace GPS-based accuracy evaluation, but to complement it by addressing limitations that accuracy alone cannot capture. Real geolocalization use cases often involve multi-granularity intent, where users may care more about correct reasoning and region-level understanding than exact coordinates. Distance-based metrics cannot reflect such reasoning quality or the intermediate spatial choices a model makes.
>
> We also examined whether user preference diverges from true geolocalization quality. In Section 4.3, expert annotators independently evaluated sampled pairs. According to prior study [1], we observed strong agreement between crowd votes and expert choices, indicating that user preference is not dominated by superficial features.
>
> Thus, the leaderboard does not aim to show “which model tricks users”, but to provide a complementary signal about reasoning quality and human-aligned geolocalization behavior, which we believe is meaningful for the research community.
>
> > GeoArena-1K is composed of images uploaded by users and their preference response. Can this data be publicly released? Have the users been asked for consent? I could not find a link to GeoArena in the paper hence I couldn't try it myself to see if there is a button to consent to the release of my preference data and uploaded images. If not, that data can't be released.
>
> Thank you for the question. We confirm that all users contributing to GeoArena-1K were asked for consent before submitting any images or preference votes. The online platform includes a clear consent statement explaining that uploaded images and voting data may be used for research purposes and released in anonymized form. Due to the double-blind requirement of the ICLR submission, we were unable to include the platform URL in the paper, but GeoArena is already publicly accessible. In the revised version, we also open-source our code base to support other projects in building similar platforms. Please refer to the abstract section. Additionally, we have included a new section in the Appendix (Appendix A.6) to provide a detailed discussion of user consent.

---

> ### Author Response · Authors · 2025-11-20
> **Response [2/2]**
>
> > The prompt is provided by the user, meaning that the user could easily ask anything about an image. Is there anything that ensures that the platform is constrained to geolocalization, or can the platform be used for any kind of image-text "battle" between two VLMs? Line 049 says that a limitation of GPS-based evaluation is that existing evaluations only consider the spatial distance between the final prediction and the ground truth, ignoring the model’s reasoning process. Why is that a limitation? In most cases the user would only care about having an accurate geolocalization
>
> Thank you for the thoughtful questions. Regarding the first point, we intentionally do not restrict the user’s prompt to predefined templates, because real-world geolocalization needs are diverse and often require different levels of detail or contextual reasoning. To ensure the reliability of the leaderboard, as described in Section 3.2, we manually filter inappropriate or irrelevant submissions.
>
> Inspired by your comments, we explored using LLMs as a judge to filter inputs.
>
> We sampled 100 prompts from our voting data and labeled them as True (indicating they inquire about geolocating an image). We also sampled 100 prompts from the Chatbot Arena dataset [2] and labeled them as False. We then asked several LLMs (Gemini 2.0 Flash, GPT-3.5-Turbo, and GPT-4.1-Mini) to classify each prompt using the instruction:
>
> ```
> You are a prompt classifier. Analyze the provided user prompt and determine if it is asking about image geolocalization.
>
> Image geolocalization refers to determining or estimating the geographic location (e.g., city, country, landmark) where an image was taken based on its visual content.
>
> Return ONLY a JSON object with one key: "is_geo".
> The value must be "true" if the prompt is inquiring about geolocalizing an image (e.g., "Where was this photo taken?" or instructions for an expert in image geolocalization), or "false" otherwise.
> If uncertain, default to "false".
>
> Output format (no extra words):
> {{"is_geo": "true"|"false"}}
>
> User prompt: {user_prompt}
> ```
>
> All three models achieved 100% accuracy. This is largely because most users adopt the default prompt provided by GeoArena, and prompts that request geolocation exhibit clear and recognizable linguistic patterns. These findings suggest that we can reliably leverage large language models to filter user inputs, ensuring that the leaderboard remains focused on geolocalization queries and therefore more trustworthy.
>
> For the second point, our motivation for going beyond distance-based accuracy stems from two observations. First, the reasoning process is an essential part of geolocalization. Many users rely on the model’s intermediate explanations to judge whether a prediction is trustworthy and interpretable, and distance alone cannot capture this aspect. Second, recent work in agent-style geolocalization [3] shows that LVLMs often generate several plausible candidate regions during multi-step reasoning. In such cases, the final coordinate may be far from the correct location due to a single decision error at the final step, even if all intermediate steps were geographically coherent. Pure distance-based scoring assigns these cases an extremely low score, which does not reflect their actual reasoning quality.
>
> GeoArena therefore complements traditional accuracy metrics by evaluating human-aligned reasoning quality, rather than replacing GPS-based evaluation. We believe combining both perspectives offers a more complete understanding of model capability.
>
> [1] Chiang, Wei-Lin, et al. "Chatbot arena: An open platform for evaluating llms by human preference." *Forty-first International Conference on Machine Learning*. 2024.
>
> [2] https://huggingface.co/datasets/lmsys/chatbot_arena_conversations
>
> [3] Li, Ling, et al. "Recognition through Reasoning: Reinforcing Image Geo-localization with Large Vision-Language Models.", NeurIPS 2025.

---

> > ### Comment · Reviewer_ERsX · 2025-11-24
> > **Response to rebuttal**
> >
> > Despite the rebuttal I still believe the contribution of GeoArena is not clear. I see the point of evaluating VLMs/LLMs for helpfulness (like LMArena) or evaluating VLM geolocation capabilities, but evaluating something in between (like GeoArena) is in my opinion not a useful contribution to the community

---

> > > ### Author Response · Authors · 2025-11-26
> > > **Response**
> > >
> > > We appreciate the reviewer’s insightful feedback. We understand the concern that GeoArena might seem like an "in-between" metric. However, we respectfully argue that human preference is not an intermediate state, but an orthogonal and necessary dimension for evaluating generative intelligence, parallel to recent shifts in other domains.
> > >
> > > **Analogy to Image Generation (GenAI-Arena [1]):** Consider the evaluation of text-to-image models. While quantitative metrics like FID or CLIP Score exist (analogous to "distance error" in geolocation), they often fail to capture the nuanced quality and user satisfaction associated with generative outputs. The community introduced platforms like GenAI-Arena because human evaluation provides the "ground truth" for generation quality that metrics miss. Similarly, in geolocalization, a coordinate distance metric captures where the prediction is, but fails to capture **why** and **how** the model reached it.
> > >
> > > **Utility for Future Training:** The utility of GeoArena extends beyond a leaderboard. To train models that truly understand geography (rather than memorizing coordinates), we need preference data to align models via RLHF. Distance loss alone is insufficient for training reasoning chains. GeoArena provides the community with the first large-scale human preference dataset in this domain, enabling research into reward modeling for geographic reasoning.
> > >
> > > We hope these clarifications better articulate our motivation and the distinct contribution of GeoArena to the research community. We sincerely appreciate your time and support.
> > >
> > > [1] Jiang, Dongfu, et al. "Genai arena: An open evaluation platform for generative models." Advances in Neural Information Processing Systems 37 (2024): 79889-79908.

---

### Official Review · Reviewer_zGCV · 2025-10-30

**Soundness:** 2
**Presentation:** 3
**Contribution:** 2
**Rating:** 2
**Confidence:** 3

**Summary:**

This paper proposes a dynamic, user-preference-based benchmarking platform for image geolocalization. The platform allows users to upload their own images, provide feedback, and contribute to the evaluation of vision-language models (VLMs) on geolocation tasks.

**Strengths:**

1. The paper is clearly written and well-structured, making it easy to follow.

2. The image geolocation task is an interesting and practically relevant problem. Leveraging pretrained VLMs for this purpose is a natural and meaningful direction, and benchmarking diverse VLMs on this task is important.

3. The proposed platform mitigates data leakage issues in VLM evaluation by enabling users to contribute their own evaluation data, thus creating a more dynamic and realistic benchmark.

4. The authors have implemented and deployed the platform, and the system and models are open-sourced, which enhances reproducibility and potential community impact.

**Weaknesses:**

1. Engineering-focused contribution.

The paper’s main contribution appears to be the development of a platform for data collection and do the evaluation based on the collected data. As such, the work leans more toward an engineering or system contribution than an academic one.

2. Granularity issues.

The user-contributed data may vary in granularity. For example, some users may specify only the country, while others may provide exact coordinates or city names. Without a clear mechanism to handle this variation, the benchmark may not produce consistent or fine-grained evaluations across samples.

3. Relevance and data quality concerns.

While user participation helps prevent data leakage, it introduces potential quality and relevance issues. Some users may submit irrelevant images, low-quality inputs, or noisy feedback, which can make the rankings unreliable or manipulable by unintentional or malicious contributions.

4. Safety and privacy risks.

Allowing open user contributions raises safety and privacy concerns. For instance, attackers might upload unsafe or inappropriate content to manipulate model rankings or compromise platform integrity. The paper would benefit from a clearer discussion of content moderation and safety mechanisms.

5. Volume and diversity dependence.

The benchmark’s effectiveness depends heavily on user participation and data diversity. If only a small or geographically concentrated group of users contributes, the resulting rankings could be biased or unrepresentative of global performance.


I appreciate the authors’ efforts to design a more practical and dynamic benchmarking platform for VLMs. However, several practical challenges, including data quality, safety, and the scientific novelty of the contribution, remain insufficiently addressed. Clarifying how these issues are mitigated would significantly strengthen the paper’s impact and credibility.

**Questions:**

Please see above.

---

> ### Author Response · Authors · 2025-11-20
> **Response [1/2]**
>
> > W1: Engineering-focused contribution.
> > The paper’s main contribution appears to be the development of a platform for data collection and do the evaluation based on the collected data. As such, the work leans more toward an engineering or system contribution than an academic one.
>
> Thank you for the comment. We understand the concern that our work may seem mainly engineering-oriented. We have revised the paper to clarify that the platform is only one realization of a broader research goal: addressing long-standing issues in geolocalization evaluation, including static-dataset leakage, missing reasoning assessment, and privacy concerns. Our main contribution is proposing the first dynamic, user-preference-based evaluation framework for this task, along with the associated analysis and dataset. We have revised our contributions to emphasize these motivations more clearly in the updated version.
>
> > W2: Granularity issues.
> > The user-contributed data may vary in granularity. For example, some users may specify only the country, while others may provide exact coordinates or city names. Without a clear mechanism to handle this variation, the benchmark may not produce consistent or fine-grained evaluations across samples.
>
> Thank you for the insightful comment. We agree that user-contributed data naturally vary in granularity. In our design, we intentionally avoid enforcing fixed granular categories because our goal is to build a unified platform that reflects model performance across all levels of geographic specificity. Introducing strict granularity constraints could lead to an endless set of subdivisions (e.g., region-level, language-specific regions, or other fine distinctions). That said, we agree this is an interesting direction, and future extensions could explore granularity-aware evaluation protocols. Thank you again for the insights here!
>
> > W3: Relevance and data quality concerns.
> > While user participation helps prevent data leakage, it introduces potential quality and relevance issues. Some users may submit irrelevant images, low-quality inputs, or noisy feedback, which can make the rankings unreliable or manipulable by unintentional or malicious contributions.
>
> Thank you for raising this concern. As described in Section 3.2, we filter out inappropriate or sensitive inputs  manually to ensure the reliability of the leaderboard. This manual process is feasible in our current scale. Following your suggestion, we agree that an automated solution would be beneficial, and we conducted an additional experiment using a VLM as an automatic filter.
>
> We sampled 100 prompts from our voting data and labeled them as True (indicating they inquire about geolocating an image). We also sampled 100 prompts from the Chatbot Arena dataset [1] and labeled them as False. We then asked several LLMs (Gemini 2.0 Flash, GPT-3.5-Turbo, and GPT-4.1-Mini) to classify each prompt using the instruction:
>
> ```
> You are a prompt classifier. Analyze the provided user prompt and determine if it is asking about image geolocalization.
>
> Image geolocalization refers to determining or estimating the geographic location (e.g., city, country, landmark) where an image was taken based on its visual content.
>
> Return ONLY a JSON object with one key: "is_geo".
> The value must be "true" if the prompt is inquiring about geolocalizing an image (e.g., "Where was this photo taken?" or instructions for an expert in image geolocalization), or "false" otherwise.
> If uncertain, default to "false".
>
> Output format (no extra words):
> {{"is_geo": "true"|"false"}}
>
> User prompt: {user_prompt}
> ```
> All three models achieved 100% accuracy. This is largely because most users adopt the default prompt provided by GeoArena, and prompts that request geolocation exhibit clear and recognizable linguistic patterns. These findings suggest that we can reliably leverage large language models to filter user inputs, ensuring that the leaderboard remains focused on geolocalization queries and therefore more trustworthy.
>
> > W4: Safety and privacy risks.
> > Allowing open user contributions raises safety and privacy concerns. For instance, attackers might upload unsafe or inappropriate content to manipulate model rankings or compromise platform integrity. The paper would benefit from a clearer discussion of content moderation and safety mechanisms.
>
> Thank you for the suggestion. As noted in our response to W3, we currently rely on manual filtering to remove unsafe, inappropriate, or low-quality submissions, which is practical at our present scale. We have also explored using an LLM as an automatic content filter, and the corresponding results are included in the updated version. All content is included in the updated version (Appendix A.5).

---

> ### Author Response · Authors · 2025-11-20
> **Response [2/2]**
>
> > Volume and diversity dependence.
> > The benchmark’s effectiveness depends heavily on user participation and data diversity. If only a small or geographically concentrated group of users contributes, the resulting rankings could be biased or unrepresentative of global performance.
>
> Thank you for raising this concern. We agree that the volume and geographic spread of user submissions naturally influence the data distribution. As an open platform, GeoArena does not restrict who can participate, and we believe this openness is essential to reflecting real use cases rather than enforcing artificial sampling constraints. Since GeoArena is a live open platform, we do not restrict where users come from. As the platform continues to attract a broader user base over time, the geographic bias in contributions is expected to naturally decrease.
>
> [1] https://huggingface.co/datasets/lmsys/chatbot_arena_conversations

---

> > ### Comment · Reviewer_zGCV · 2025-11-25
> > **Reply to the rebuttal**
> >
> > Thanks to the authors for the detailed rebuttal. However, I am still concerned about the novelty of the proposed approach and the trade-offs between the drawbacks of an open platform and the advantages it aims to provide. Although the authors demonstrate that a VLM can be used as an automatic filter for privacy and safety issues, the method appears rather naïve and is unable to filter out certain images that may reveal private locations or pose safety risks. For instance, a malicious user could repeatedly provide incorrect feedback to mislead the model.
> >
> > While the open platform indeed mitigates some issues such as static test leakage by allowing broad participation, it simultaneously introduces additional challenges. These include potential bias and lack of diversity in the collected data (even though the authors argue that a broader user base may emerge over time, this assumption relies on worldwide participation, which seems unrealistic), as well as concerns regarding privacy, safety, and the difficulty of controlling data granularity.
> >
> > Given these concerns, it is difficult to conclude that the advantages outweigh the disadvantages. Therefore, I have decided to maintain my original score.

---

### Author Response · Authors · 2025-11-20
**General Response**

We sincerely thank the AC and all reviewers for their thoughtful feedback. Our work addresses a central challenge in image geolocalization: current evaluation protocols rely almost entirely on static datasets and distance-based accuracy, which do not reflect how humans assess geolocalization quality in practice. Existing datasets risk overlap with pretraining sources, lack the ability to measure reasoning quality, and do not reflect the open-world and multi-granularity nature of real user needs. GeoArena is designed to fill this gap by providing the first dynamic, user-preference-based evaluation framework for geolocalization.

---

In response to reviewer comments, we have updated the paper to clarify our motivation, strengthen the presentation of our contributions, and incorporate several improvements described below.

**1. We open-sourced our code base via an anonymous link**, now included in the abstract.

**2. We added a new experiment evaluating whether LLMs can automatically filter user prompts**, included in **Appendix A.5**.

**3. We revised our contribution section to more clearly highlight our motivations and conceptual advances.**

**4. We added a new subsection on user consent**, included in **Appendix A.6**.

We thank the reviewers again for their constructive feedback. We believe the revisions address all major concerns and strengthen both the clarity and the impact of the paper.

---

### Note · Authors · 2026-01-02

I have read and agree with the venue's withdrawal policy on behalf of myself and my co-authors.